# Loaded inter-set stretch may selectively enhance muscular adaptations of the plantar flexors

**Derrick W. Van Every**[1], **Max Coleman**[1], **Avery Rosa**[1], **Hugo Zambrano**[1], **Daniel Plotkin**[1], **Xavier Torres**[1], **Mariella Mercado**[1], **Eduardo O. De Souza**[2], **Andrew Alto**[1], **Douglas J. Oberlin**[1], **Andrew D. Vigotsky**[3], **Brad J. Schoenfeld**[1] *

**1** Department of Health Sciences, CUNY Lehman College, Bronx, NY, United States of America,
**2** Department of Health Sciences & Human Performance, The University of Tampa, Tampa, FL, United States of America, **3** Departments of Biomedical Engineering and Statistics, Northwestern University, Evanston, IL, United States of America

* brad@workout911.com

**Data Availability Statement:** The data for this study are available from the OSF preregistration page (https://osf.io/mtw6q).

## Abstract

The purpose of this study was to evaluate differences in changes in muscle strength and muscle thickness (MT) of the plantar flexor muscles between traditional resistance training (RT) involving passive rest and RT combined with inter-set stretch in the calf raise exercise. Employing a within-subject design, 21 young, healthy men performed plantar flexion exercises twice per week in both a traditional RT (TRAD) format and combined with a 20-second inter-set stretch (STRETCH). One leg was randomly assigned to the TRAD condition and the contralateral leg performed the STRETCH condition throughout the 8-week study period. Dependent variables included MT of the lateral gastrocnemius (LG), medial gastrocnemius (MG) and the soleus (SOL), and isometric strength of the plantar flexors. Results indicated a potential beneficial hypertrophic effect of STRETCH compared to TRAD for the SOL [0.7 mm, $CI_{90\%} = (0, 1.6)$], while the LG had more ambiguous effects [0.4 mm (−0.4, 1.3)] and MG effects were equivocal [0 mm (−0.6, 0.7)]. In general, LG demonstrated greater standardized growth [$z = 1.1$ (1, 1.3)] as compared to MG [$z = 0.3$ (0.2, 0.5)] and SOL [$z = 0.3$ (0.2, 0.5)]. Measures of isometric strength showed a modest advantage to STRETCH. In conclusion, loaded inter-set stretch may enhance MT of the soleus but effects on the gastrocnemii appear uncertain or unlikely in untrained men; plantar flexor strength appears to be modestly enhanced by the interventional strategy.

## Introduction

Stretching a muscle under loaded conditions promotes significant muscular adaptations in animal models when applied consistently over time. Notably, seminal research in quails found muscle mass increases of >50% in the anterior latissimus dorsi after just several weeks of loaded stretch [1–3]. However, these protocols involve extreme interventions whereby a Velcro tube filled with lead pellets is wrapped around the birds' stretched wings for days on end.

**Funding:** This study was supported by a PSC CUNY grant from the State of New York received by BJS. Award # 63497-00 51. The funding agency played no role in study design, data collection, analysis, decision to publish, or preparation of the manuscript. There was no additional funding received for this study.

**Competing interests:** I have read the journal's policy and the authors of this manuscript have the following competing interests: BJS serves on the scientific advisory board of Tonal Corporation, a manufacturer of exercise equipment. Tonal did not provide any funding for the study, nor were they involved in the design, methods, analysis or write-up of the manuscript. This competing interest does not alter our adherence to PLOS ONE policies on sharing data and materials.

Thus, findings cannot necessarily be extrapolated to humans performing traditional stretching protocols, which generally include brief bouts of stretching held for 15 to 30 seconds [4].

A recent human trial reported that 6-weeks of isolated loaded stretch of the plantar flexors produced significant increases in muscle thickness (MT) [5]. It has been speculated that stretching between sets of resistance exercise may enhance human muscle hypertrophy over and above that achieved with traditional resistance training (RT) or isolated stretching alone [6]. In support of this hypothesis, Evangelista et al. [7] found a potential hypertrophic benefit to performing 30 seconds of static stretching versus passive rest between sets during regimented RT in untrained individuals. Although these results are intriguing, it should be noted that the stretch was unloaded; hence, whether adding resistance to the stretch would have enhanced results remains underdetermined. In this regard, Silva et al. [8] reported markedly greater increases in muscle thickness favoring a group of resistance trained individuals that performed loaded inter-set stretch of the calf muscles compared to a group that rested passively; however, findings were presented as a conference abstract and never published in a peer-reviewed journal, thus precluding the ability to scrutinize methods and results. If these results can be corroborated, loaded stretch would provide practitioners with an intriguing strategy to enhance muscular adaptations in a time-efficient manner.

To our knowledge, only one published study has endeavored to compare the effects of RT involving passive inter-set rest vs. RT that included a loaded stretch protocol between sets. Wadhi et al. [9] randomized resistance-trained men to perform bench press exercises with either a 30-second loaded stretch between sets or a passive inter-set rest. Training was carried out twice per week for 8 weeks. Results showed similar increases in muscle thickness of the pectoralis major between conditions. It should be noted that the inter-set stretch was performed with 15% of their working load from the prior set on a different exercise (cable fly), which may not have imposed a sufficient stimulus to augment hypertrophy. Moreover, given the preliminary findings of Silva et al. [8], it is possible that loaded stretch may confer differential effects on muscular adaptations between the upper and lower body musculature, or perhaps muscles of different architectures and fiber type composition.

Given the paucity of research and contradictory findings on the topic, we aimed to evaluate changes in muscle strength and MT of the calf muscles between traditional RT involving passive rest and RT combined with inter-set stretch in plantar flexion exercises. A secondary aim was to determine if the inclusion of an inter-set stretch has differential effects on MT of the individual plantar flexors (i.e., soleus versus gastrocnemii). We hypothesized that muscular adaptations would be greater in the limb performing RT combined with inter-set stretch [10], and that the lateral gastrocnemius would experience greater MT than the other plantar flexors [11].

## Materials and methods

### Participants

Twenty-five healthy, untrained but recreationally active males from a university population (height: 175.1 ± 7.0 cm; weight: 80.4 ± 19.6 kg; age: 20.8 ± 6.1 yrs; body fat: 22.7 ± 10.5%) volunteered to participate in this study. Participants had not performed regimented lower-body RT for at least 6 months prior to participation, although some reported limited previous RT experience. The sample size was justified by *a priori* precision analysis for the minimum detectable change at the 90% level (MDC$_{90\%}$) for medial gastrocnemius thickness (i.e., SEM × $z_{0.05}$ × $\sqrt{2}$ = 0.7 mm), such that the compatibility interval (CI) of the between-group effect would be approximately ± MDC$_{90\%}$. Based on data from previous research [11], along with their sampling distributions, Monte Carlo simulation was used to generate 90% CI widths for 1000 random samples of each sample size. To ensure a conservative estimate, as literature

values may not be extrapolatable, the sum of each simulated sample size's 90% CI's mean *and* SD was used, and the smallest sample that exceeded $MDC_{90\%}$ was chosen; that is, 17 participants. To account for the possibility of attrition, we recruited 25 participants.

Participants were required to meet the following inclusion criteria: 1) between the ages of 18–35; 2) no existing musculoskeletal disorders, neuromuscular disorders, lower extremity pain, or prior traumatic injury to the triceps surae/Achilles complex; 3) self-reportedly free from consumption of anabolic steroids or any other legal or illegal agents known to increase muscle size for the previous year, and; 4) had not performed regimented RT for the lower body musculature in the past 6 months.

The study employed an individually randomized within-subject design where each participant performed both traditional RT (TRAD) and RT combined with inter-set stretch (STRETCH) for the plantar flexors. One leg was randomly assigned to the TRAD condition and the contralateral leg performed the STRETCH condition throughout the study period. A within-participant design allows for increased precision of effect estimation, especially with the high pre-post measurement correlations that are observed in muscle thickness studies [12]. Random allocation as to which limb received which stimulus was carried out using block randomization, with two participants per block, in R [13]. Approval for the study was obtained from the university Institutional Review Board. Written informed consent was obtained from all participants prior to beginning the study. All training was carried out in the college fitness center and testing was performed in a laboratory setting. The methods for this study were pre-registered prior to recruitment (https://osf.io/mtw6q).

## Resistance training procedures

To ensure involvement of the entire triceps surae musculature [14], the RT protocol consisted of the seated and straight-leg calf raise exercises, performed for 2 weekly sessions on non-consecutive days. The seated calf raise was carried out on a plate-loaded unit (Body Masters, Rayne, LA) and the straight-leg calf raise was carried out on a leg press machine (Life Fitness, Franklin Park, IL). Training times within each participant were consistent across the duration of the study but varied between participants to allow the protocol to fit within each participant's schedule. A one-week familiarization period was provided prior to the study whereby participants performed these exercises unilaterally over 3 non-consecutive days using their bodyweight and the raw weight of the machine for 3 sets of 5, 10, and 15 repetitions per set on Days 1, 2, and 3, respectively. This was done to promote a repeated bout effect and thus help prevent excessive muscle soreness from interfering with training during the early stages of the training phase [15].

Prior to the training phase, all participants underwent repetition maximum (RM) testing for their 10RM on both the seated and straight-leg calf exercises to determine individual initial training loads. The RM testing was consistent with recognized guidelines as established by the National Strength and Conditioning Association [16]. Thereafter, participants engaged in 8 weeks of intensive training of the plantar flexors, during which the two interventions were provided concurrently. To minimize any potential confounding effects from exercise order, TRAD was performed first in Session 1, STRETCH was performed first in Session 2, and then the conditions continued alternating in this fashion for the duration of the study period.

Participants performed 4 sets per exercise per session with 2-minutes of rest afforded between sets and ~3 minutes of rest afforded between exercises. For STRETCH, participants descended into a loaded stretch immediately following completion of each set using the same load employed during the set. The stretch was held for 20 seconds, and then subjects rested passively for the remaining duration of the rest interval (i.e., a total of 100 seconds rest between

each set). Alternatively, TRAD rested passively throughout the duration of each rest interval. Thus, the total time between sets remained identical for each condition. Sets were carried out to the point of momentary concentric muscular failure—herein defined as the inability to perform another concentric repetition while maintaining proper form—with a target repetition range of 8 to 12RM. The load was adjusted for each exercise as needed on successive sets to ensure that participants achieved failure within the target repetition range. Cadence of repetitions was carried out with a ~1 second concentric action and a ~2-second eccentric action. All routines were directly supervised by research assistants experienced with RT to ensure proper performance of the respective routines. Loads were progressively increased each week within the confines of maintaining the target repetition range for each condition. Participants were instructed to refrain from performing any additional resistance-type lower body training for the duration of the study. A timeline of the study is displayed in Fig 1.

## Measurements

**Anthropometry.** Baseline anthropometric data were collected on the initial visit to the laboratory. Participants were instructed to refrain from eating for at least 8 hours prior to testing, eliminate alcohol consumption for 24 hours, abstain from strenuous exercise for 24 hours, and void immediately before the test. Participants' height was measured to the nearest 0.1 cm using a stadiometer; weight was assessed to the nearest 0.1 kg on a calibrated scale (InBody 770; Biospace Co. Ltd., Seoul, Korea).

**Muscle thickness.** Ultrasound imaging was used to obtain measurements of muscle thickness (MT) of the medial gastrocnemius (MG), lateral gastrocnemius (LG), and soleus (SOL) with participants lying prone and ankles maintained in neutral position. A trained sonographer performed all testing using a B-mode ultrasound imaging unit (Sonoscape E1; Shenzhen, China). The technician, who was blinded to limb allocation, applied a water-soluble transmission gel (Aquasonic 100 Ultrasound Transmission gel, Parker Laboratories Inc., Fairfield, NJ) to each measurement site, and a 4–10 MHz ultrasound probe was placed perpendicular to the tissue interface without depressing the skin. Measurements for each respective site were taken with a tape measure on the posterior surface of both legs at 30% of the lower leg length (the distance from the articular cleft between the femur and tibia condyles to the lateral malleolus) on the medial and lateral sides, which were marked with a felt-tip pen to ensure consistency of measures. When the quality of the image was deemed to be satisfactory, the technician saved the image to hard drive and obtained MT dimensions for the MG, LG, and SOL using the machine's calculation package. Fig 2 displays an example of an ultrasound image obtained from one of the participants.

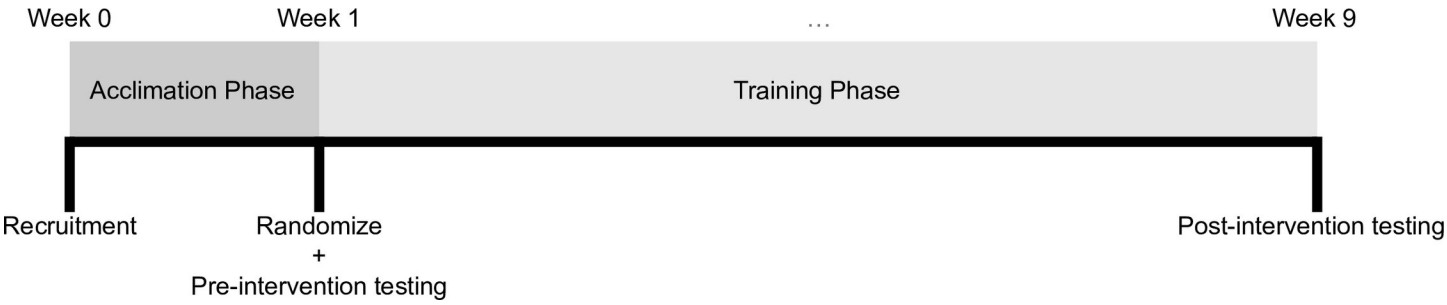

**Fig 1. Study timeline.** After being accepted into the study, all participants went through a 1-week acclimation phase. Thereafter, the limbs of participants were randomized to their respective conditions and underwent preintervention testing. The intervention lasted 8 weeks, after which participants underwent postintervention testing.

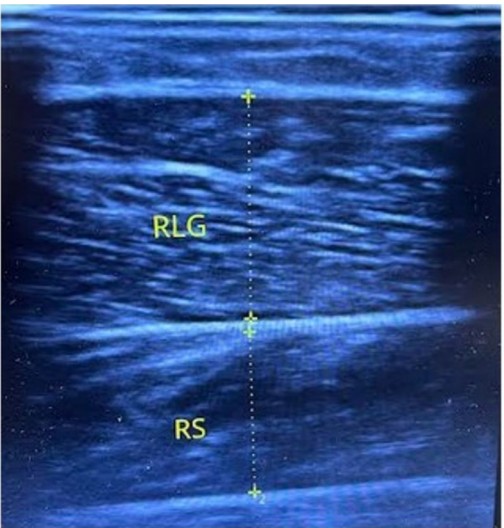

**Fig 2. Example of an ultrasound image: A representative image illustrating the measurement of muscle thickness.**

Muscle thickness of the MG and LG was determined as the distance from the superficial to deep aponeuroses that borders the SOL. The SOL was measured from the upper and lower aponeuroses separating the muscle. In an effort to ensure that swelling in the muscles from training did not confound results, images were obtained ≥48 hours after the acclimation phase, as well as ≥ 48 hours after the final training session. This is consistent with research showing that acute increases in MT return to baseline within 48 hours following a RT session [17] and that muscle damage is minimal after repeated exposure to the same exercise stimulus over time [18]. To further ensure accuracy of measurements, 3 images were obtained for each site and their averages were used as the final value for MT. The intraclass correlation coefficients (ICC) from our lab for the MG, LG, and SOL are 0.990, 0.993, and 0.990, with corresponding standard errors of the measurement (SEM) of 0.44, 0.58 and 0.82 mm, respectively.

## Maximal strength assessments

**Muscle strength.** To test isometric ankle plantar flexion strength, each participant was secured in a dynamometer (Biodex Isokinetic Dynamometer System 4 Pro, Shirley, NY) with their hips positioned to 85˚ flexion and testing ankle to 90˚ (i.e., foot 90˚ relative to the tibia). Participants were instructed to extend their leg as forcefully as possible against the machine pad and were then given a practice trial prior to testing for acclimation with the test. Testing was performed with two different knee-joint positions: full extension (0˚) and 90˚ flexion. Knee extension facilitates force production from the LG and MG, in addition to the SOL. Conversely, knee flexion helps to isolate the SOL by placing the LG and MG under active insufficiency [19]; therefore, the difference in net joint moments between the two conditions can be considered the contribution from the gastrocnemii. Each trial consisted of a maximum voluntary isometric effort, which lasted for 5 seconds, and was followed by a 30-second rest interval. A total of 4 trials were performed for each knee-joint position. Participants were verbally encouraged throughout each trial and were allowed to view the screen for biofeedback and increased performance [20]. The highest peak net joint moment from each of the 4 trials for each position was used for analysis. The ICCs from our lab for the isometric plantar flexion strength test with legs straight and in 90˚ flexion are 0.776 and 0.809, with corresponding SEMs of 12.4 and 20.0 N·m, respectively.

## Statistical analyses

To assess the differential effects of TRAD versus STRETCH, all data were analyzed in R (version 4.1.1), in which hierarchical linear models (HLM) were constructed [13, 21]. HLMs are similar to linear mixed-effects models, with which readers may be more familiar, but are conceptualized differently since specified in terms of "levels." HLMs or mixed-effects models are indicated here because of our within-participant design. A single model was constructed to obtain two effects, which followed the form:

*Level 1*

$$post_{ij} = \beta_{0j} + \beta_{1j}(pre_{ij}) + \beta_{2j}(intervention_{ij}) + \epsilon_{ij}$$

*Level 2*

$$\beta_{oj} = \gamma_{00} + r_{0j}$$
$$\beta_{1j} = \gamma_{10}$$
$$\beta_{2j} = \gamma_{20}$$

for participant $i$ and limb $j$, where level 1 is hypertrophy (within-participant), level 2 is between-participant, and $\beta_{2j}$ is the effect of interest, which is the estimate of the differential effect of the intervention on a muscle (e.g., on the medial gastrocnemius). This was estimated separately for MG, LG, and SOL. $intervention_{ij}$ was dummy-coded 0 for TRAD and 1 for STRETCH; $pre_{ij}$ was group mean- (i.e., participant mean-) centered. From a linear mixed-effects modeling perspective, this is simply a model with random intercepts for each participant.

Secondary analyses were carried out to assess within-position strength adaptations. For each analysis, post-intervention score was the dependent variable, intervention (i.e., STRETCH or TRAD) was the independent variable, pre-intervention scores were a covariate, and there were varied intercepts for each participant so that all analyses are within-participant, such that the hierarchical linear model took the following form:

*Level 1*

$$post_{ij} = \beta_{0j} + \beta_{1j}(pre_{ij}) + \beta_{2j}(intervention_{ij}) + \epsilon_{ij}$$

*Level 2*

$$\beta_{oj} = \gamma_{00} + r_{0j}$$
$$\beta_{1j} = \gamma_{10}$$
$$\beta_{2j} = \gamma_{20}$$

where level 1 is strength or hypertrophy (within-participant), level 2 is between-participant, and $\beta_{2j}$ is the effect of interest.

Finally, we explored the differences in muscle growth between the muscles by $z$-scoring each muscle using its pre-intervention scores, and we used these in the hierarchical linear

model with muscle interactions,

*Level 1*

$$post_{ij}^z - pre_{ij}^z = \beta_{0j} + \beta_{1j}(pre_{ij}^z) + \beta_{2j}(intervention_{ij}) + \beta_{3j}(MG_{ij}) + \beta_{4j}(LG_{ij})$$

$$+ \beta_{5j}(MG_{ij}pre_{ij}^z) + \beta_{6j}(LG_{ij}pre_{ij}^z) + \beta_{7j}(MG_{ij}intervention_{ij})$$

$$+ \beta_{8j}(LG_{ij}intervention_{ij}) + \epsilon_{ij}$$

*Level 2*

$$\beta_{0j} = \gamma_{00} + r_{0j}$$

$$\beta_{[1-8]j} = \gamma_{[1-8]0}.$$

Estimated marginal means were used to obtain the standardized baseline-adjusted change scores, their contrasts, and the 90% CIs using Kenward-Roger degrees of freedom [22]. This model was not pre-registered, but rather was *post hoc* and used to help contextualize our findings.

For all analyses, the bootstrap was used to obtain bias-corrected and accelerated 90% compatibility intervals (CI) of the point estimate of each effect. We analyzed data per-protocol rather than intention-to-treat since we were interested in the effect of the intervention rather than its prescription. Finally, to avoid dichotomous interpretations of the results, no a priori α-level was set. Rather than interpreting effects from a single test, or set of tests, the results were interpreted on a continuum using all statistical outcomes, in combination with theory and practical considerations [23, 24].

## Results and discussion

Of the 25 participants that initially volunteered for participation, 4 dropped out of the study for the following reasons: non-compliance ($n = 2$), injury unrelated to the study ($n = 1$), and non-musculoskeletal adverse event experienced during the study ($n = 1$). Thus, 21 participants completed the entire study protocol, which indicates good statistical precision based on our *a priori* precision analysis. All participants attended >90% of the RT sessions. Fig 3 displays a CONSORT diagram of the data collection process.

### Effect of inter-set stretching on muscle thickness

The LG and SOL had modest estimated effects of STRETCH relative to TRAD, with point estimates of 0.4 mm and 0.7 mm, respectively. However, we observed variability associated with these estimates. For the LG, our data are compatible with values ranging from −0.4 mm (favoring TRAD) to 1.3 mm (favoring STRETCH). For the SOL, our data are compatible with values ranging from 0 to 1.6 mm (favoring STRETCH). In contrast, results for the MG were equivocal—our point estimate was zero and the data were compatible with estimates ranging from −0.6 mm (favoring TRAD) to 0.7 mm (favoring STRETCH) (Table 1 and Fig 4A).

### Between-muscle growth

We *z*-scored each muscle to compare growth between muscles. Marginalizing over condition, LG demonstrated greater growth than MG [$z = 0.8$ (0.6, 1)] and SOL [$z = 0.8$ (0.6, 1)], and the difference between MG and SOL was equivocal with appreciable variance [MG minus SOL: $z = 0$ (−0.2, 0.3)]. Marginally, LG growth was $z = 1.1$ (1, 1.3), while MG and SOL were both $z = 0.3$ (0.2, 0.5). Differential effects of the intervention were estimated to be small but with

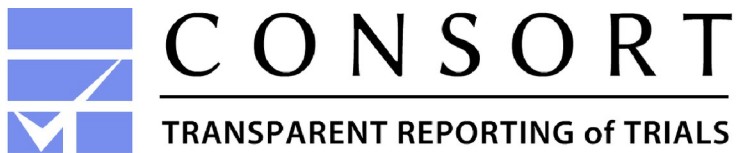

## CONSORT 2010 Flow Diagram

**Enrollment**

Assessed for eligibility (n=29)

Excluded (n=4)
- ♦ Not meeting inclusion criteria (n=3)
- ♦ Declined to participate (n=1)

Randomized (n=25)

**Allocation**

Limb allocated to STRETCH (n=25)

Limb allocated to TRAD (n=25)

**Follow-Up**

Discontinued intervention (give reasons) (n=4)
- ♦ Non-compliance (n=2)
- ♦ Injury unrelated to study (n=1)
- ♦ Adverse event (n=1)

Discontinued intervention (give reasons) (n=4)
- ♦ Non-compliance (n=2)
- ♦ Injury unrelated to study (n=1)
- ♦ Adverse event (n=1)

**Analysis**

Analysed (n=21)
- ♦ Excluded from analysis (n=0)

Analysed (n=21)
- ♦ Excluded from analysis (n=0)

**Fig 3. Consort diagram.** Flow chart illustrating the data collection process.

poor precision [SOL minus MG: $z = 0.2$ ($-0.3$, $0.7$); SOL minus LG: $z = 0.1$ ($-0.4$, $0.6$); MG minus LG: $-0.1$ ($-0.6$, $0.4$)].

## Effect of inter-set stretching on isometric plantar flexion strength

Point estimates indicated that isometric plantar flexion strength increases modestly favored STRETCH relative to TRAD. However, CIs indicate the data were also compatible with negligible (~0) to potentially meaningful effects ($\geqslant$10% of the baseline strength) (Table 1 and Fig 4B).

Our study produced several novel findings that help to fill important gaps in the current literature on the strength- and hypertrophy-related effects of loaded inter-set stretch. Below we discuss these findings in the context of available evidence, as well as speculating on their implications for exercise prescription.

## Muscle growth between conditions

Multiple lines of evidence indicate that stretch training may elicit increases in muscle mass. Notably, research in animal models shows marked hypertrophy following relatively brief longitudinal periods of passive [25] and loaded stretch [1–3]. Moreover, high volume static stretching has been shown to elicit growth of the gastrocnemii in both passive [26] and loaded [5] conditions. Some evidence suggests that integrating stretch training into RT protocols can enhance muscle development. Evangelista et al. [7] found that the summed increases in MT for muscles of the upper and lower limbs were greater for a group of untrained individuals who performed a 30-second inter-set unloaded static stretching regimen versus a group that rested passively between sets (10.5% vs 6.7%, respectively). A conference abstract by Silva et al. [8] reported that the inclusion of a 30-second loaded stretch between sets of the straight-leg calf raise exercise produced a greater than two-fold absolute mean increase in MT compared to performing sets with passive inter-set rest periods (2.3 vs. 0.9 mm, respectively) in resistance-trained individuals. Alternatively, Wadhi et al. [9] reported that the inclusion of a 30-second loaded stretch between sets of bench press exercise did not enhance MT of the pectoralis major in resistance-trained men.

Our findings add to the body of literature on the topic, providing further insights into the potential effects of loaded inter-set stretch on muscle growth. Contrary to the findings of Silva et al. [8], we did not observe appreciable, consistent hypertrophic benefits to loaded stretch for the gastrocnemii. The MG point estimate was zero with a 90% CI that did not include appreciable effects; however, although the LG had a modest point estimate, its CI encapsulated effects ranging from relatively small negative effects to appreciable positive effects for STRETCH. The study by Silva et al. [8] was presented as a conference abstract and never published in a peer-reviewed journal, precluding our ability to reconcile discrepancies between studies. Likewise, our findings for the gastrocnemii complement those of Wadhi et al. [9], who found no benefit to loaded inter-set stretch on MT of the pectoralis major.

**Table 1. Muscle size and strength outcomes.**

| | Traditional (mean ± SD) | | Stretch (mean ± SD) | | Between-condition effect estimate (CI$_{90\%}$) |
|---|---|---|---|---|---|
| | Pre | Post | Pre | Post | |
| **Lateral Gastrocnemius (mm)** | 14.7 ± 3.1 | 17.7 ± 4.0 | 14.9 ± 2.2 | 18.2 ± 3.9 | 0.4 (−0.4, 1.3) |
| **Medial Gastrocnemius (mm)** | 18.8 ± 3.1 | 19.9 ± 3.5 | 19.0 ± 3.8 | 20.2 ± 3.7 | 0 (−0.6, 0.7) |
| **Soleus (mm)** | 17.0 ± 3.2 | 17.6 ± 3.2 | 17.3 ± 3.0 | 18.5 ± 3.9 | 0.7 (0, 1.6) |
| **Knee flexed isometric plantar flexion (N·m)** | 126 ± 46 | 148 ± 37 | 122 ± 42 | 155 ± 43 | 6 (0, 10) |
| **Knee extended isometric plantar flexion (N·m)** | 128 ± 44 | 161 ± 37 | 132 ± 34 | 169 ± 39 | 7 (0, 20) |

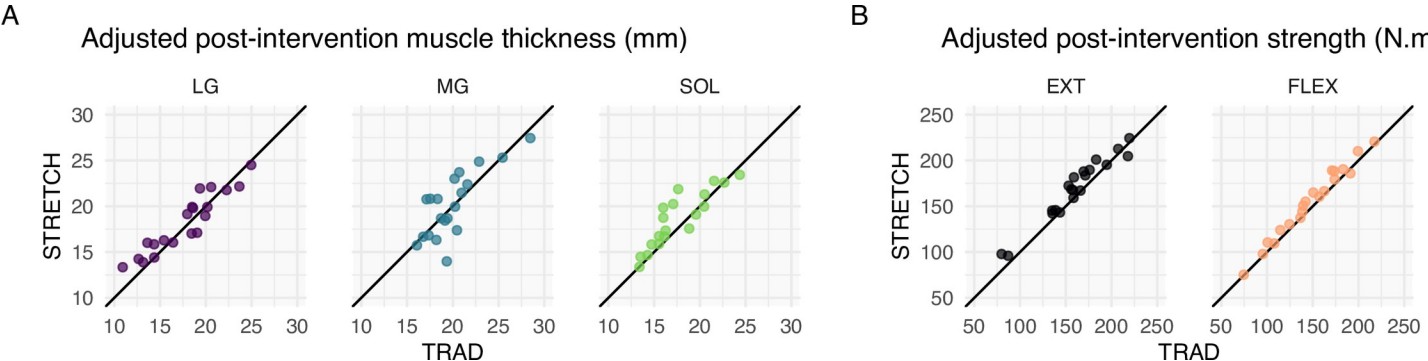

**Fig 4. Model-adjusted individual outcomes for hypertrophy and strength in the traditional (TRAD) and stretch (STRETCH) conditions.** Each data point represents an individual's model-predicted outcome for the TRAD (*x*-axis) and STRETCH (*y*-axis) conditions. The black, diagonal line is the identity line; a data point on the line indicates an identical expected outcome in TRAD and STRETCH for that individual. **(A)** Model-adjusted post-intervention outcomes in muscle thickness (mm). **(B)** Model-adjusted post-intervention outcomes in isometric plantar flexion strength (N·m). Individual lines (translucent) were demeaned within each participant and then summed with the grand mean for that exercise to stress the variability in trends rather than intercepts. Bold, opaque lines depict LOESS curves fit to the entire sample. LG = lateral gastrocnemius; MG = medial gastrocnemius; SOL = soleus; EXT = knee extended; FLEX = knee flexed.

In contrast to the gastrocnemius, our results suggest that the addition of loaded inter-set stretch in RT protocols may modestly enhance MT of the SOL. The 90% CIs around the point estimate ranged from a negligible effect to a large positive effect favoring STRETCH (0, 1.6 mm), suggesting a potentially meaningful benefit to the inclusion of loaded inter-set stretch during plantar flexion for the SOL. Although speculative, the greater time-under-tension (TUT) for STRETCH may help to explain differences between adaptations of the individual plantar flexors. Specifically, the SOL was placed under stretch for a longer duration since the gastrocnemii were actively insufficient during the seated calf raise. This may have resulted in the seated stretch preferentially targeting the SOL relative to the gastrocnemii. If the SOL's outcome can be attributed to its greater TUT, there may exist a stretch stimulus dose-response relationship that remains to be specifically investigated.

In addition to the differences in TUT owing to the uni- vs biarticular nature of the SOL and gastrocnemii, the muscles of the triceps surae also differ in other dimensions. The SOL is a tonic muscle composed predominantly of slow-twitch fibers (>80%) whereas the gastrocnemii have a relatively similar composition of both type I and type II fibers [27]. Intriguingly, research in avian models demonstrates that loaded stretch elicits greater hypertrophy in the predominantly slow-twitch latissimus dorsi compared to the patagialis, which contains a high percentage of fast-twitch fibers [28]. Whether this indicates that type I fibers are more anabolically responsive to higher TUTs compared to type II fibers or perhaps that they may have additional inherent properties predisposed to loaded stretch following performance of eccentric actions [29], such as differences in muscle architecture and geometry, remains undetermined and warrants further investigation. In particular, the SOL and gastrocnemii possess different architectural features: the SOL has shorter fibers and a shorter subtendon; the gastrocnemii have longer fibers and a longer subtendon; and the LG has a thicker subtendon than the SOL [30, 31]. These architectural features affect fiber strain and thus force production in each of the three muscles. Along these same lines, the SOL is a complex muscle with multiple, mechanically distinct compartments [32], meaning our results may not reflect hypertrophy of the whole muscle. It should be noted that the point estimate for MT of the SOL was within the SEM and results therefore should be interpreted with a degree of caution.

## Muscle growth between the plantar flexors

All of the plantar flexors analyzed showed increases in MT irrespective of the interventional condition, indicating that these muscles respond relatively well from a hypertrophy standpoint in untrained individuals. Consistent with prior research [11], the LG displayed markedly greater increases in MT compared to the MG and SOL. As previously speculated [11], the superior growth of the LG may be explained by the fact that the LG has greater fast-twitch properties compared to the other calf muscles [27], which possibly indicates a greater growth capacity [33]. Moreover, the LG is less active than the MG during standing balance and ambulation owing to its greater recruitment threshold [34, 35], suggesting it may be underutilized in novice trainees and thus have a greater hypertrophic potential during the early stages of regimented RT.

Given evidence that type II fibers have a greater growth capacity than type I fibers [33], it stands to reason that hypertrophy of the MG, a mixed-fiber muscle [27], would be superior to that of the SOL, a slow-twitch dominant muscle [27]. In support of this hypothesis, research indicates an attenuation of intracellular anabolic signaling and muscle protein synthesis in the SOL compared to muscles with inherently faster twitch properties [36, 37], as well as rodent data showing blunted hypertrophy of the SOL after synergist ablation [38, 39]. Contrarily, recent longitudinal human data found that MT increased similarly in the MG and SOL over an 8-week RT program targeting the plantar flexors [11]. The present study showed similar MT increases between the SOL and MG when results were pooled across conditions ($z = 0.3$ for both muscles). However, when considering the individual changes in MT between STRETCH and TRAD for the SOL (6.9% vs 3.5%, respectively), our findings indicate a blunted hypertrophic response for this muscle compared to the MG when performing sets in a traditional configuration; alternatively, the inclusion of inter-set stretch equalizes adaptations between the two muscles. Further mechanistic work is recommended to better understand hypertrophic adaptations of the individual human plantar flexors when subjected to RT.

## Muscle strength between conditions

We observed robust increases in measures of isometric muscle strength, both when assessed with the legs straight (pooled effect = 27.3%) and flexed at 90˚ (pooled effect = 20.2%). The point estimates modestly favored STRETCH, and the upper CI indicate the effect may be potentially meaningful; differences were somewhat more pronounced when assessing strength in a straight-leg position. The between-condition point estimates were within the SEM for both measures; thus, some caution is warranted when drawing evidence-based conclusions from these data. Our results expand on those of Wadhi et al. [9] and Silva et al. [8], who found similar increases in 1RM in the bench press and calf raise, respectively, between protocols employing traditional set configurations and sets that integrated 30-second periods of loaded inter-set stretch. Of note, our stretch measures were obtained isometrically whereas previous research on the topic involved dynamic assessment, which may help to explain discrepancies between studies. Moreover, our study included untrained men whereas the studies of Wadhi et al. [9] and Silva et al. [8] involved trained individuals. When taken as a whole, current evidence suggests minimal to modest benefits of inter-set stretch when the goal is to optimize gains in muscular strength. But also, importantly, loaded inter-set stretch does not seem to compromise muscle strength development.

Of note, baseline strength assessment showed relatively equal net plantar flexion moments between the straight and flexed positions. This finding was somewhat surprising given that the gastrocnemius is rendered actively insufficient with the knee flexed, leaving the SOL to be responsible for a majority of the moment against an imposed resistance in this position [19].

We thus would have expected that participants would have generated greater net plantar flexion moments during testing in the straight position due to the combined involvement of the LG, MG and SOL about the ankle joint, as previously demonstrated by Vigotsky et al. [19]. Discrepancies between studies may be related to differences in the dynamometers used for assessment. Namely, the dynamometer employed in the study of Vigotsky et al. [19] (Neurobionics Rotary Dynamometer) maintained subjects in a seated position with the thighs immobilized during testing whereas our unit (Biodex System 4) did not allow for immobilization of the thighs for the flexed assessment. Hence, it is feasible that participants were situated in a position more conducive to generating plantar flexion moments while their knees were flexed.

## Limitations

Our study had several limitations that must be considered when attempting to draw practical conclusions as to the implementation of loaded inter-set stretch in RT programs. First, participants were untrained young men; thus, results cannot necessarily be generalized to other populations including older individuals, women, and those with RT experience. Of note, our findings are somewhat in conflict with previous work that employed resistance-trained individuals [7, 8]; potential reasons for these discrepancies are not clear and require further investigation. Second, the experimental condition involved a 20-second stretch using the same absolute load within the confines of a 2-minute rest interval. Although anecdotally this stimulus seemed to impose a substantial challenge to the participants, it is unclear whether/how other configurations of stretch durations, rest interval lengths and/or magnitudes of load may influence results. Third, MT was measured at a single point along the length of the respective calf muscles; whether results may be different at other aspects of the plantar flexors remains undetermined. Fourth, muscle strength was assessed isometrically with a 90˚ ankle angle; results therefore cannot necessarily be extrapolated to different positions or dynamic conditions. Fifth, findings are specific to adaptations of the plantar flexors and cannot necessarily be generalized to other muscle groups. Sixth, training conditions were not necessarily the same for each participant/session, as training occurred in a commercial fitness setting, whereby research assistants were unable to control the surrounding environment during training sessions, which in turn may compromise internal validity [40]; alternatively, such conditions conceivably have a greater ecological validity than a laboratory setting and thus may provide greater insights into real-world responses. Finally, given that our study employed a within-subject design, we cannot rule out the possibility that strength-related adaptations were confounded by a cross-education effect [41]. However, it should be noted that evidentiary support for such an effect is confined to an untrained contralateral limb; it remains questionable whether cross-education occurs during longitudinal interventions where both limbs perform regimented RT. Cross-education seems to have negligible effects on hypertrophic adaptations [42], making it unlikely that the within-subject design had any influence on this outcome.

## Conclusions

Our study suggests that loaded inter-set stretch may be an effective strategy to modestly enhance MT of the SOL in young, untrained men, with unlikely appreciable hypertrophic benefits to the gastrocnemii. Given that the SOL is generally considered less responsive to anabolism compared to other skeletal muscles [36, 37], the inclusion of inter-set stretch may warrant consideration in RT programs targeting the development of this muscle. Moreover, loaded inter-set stretch appears to modestly enhance strength gains in the plantar flexors. Importantly, beneficial effects were achieved without altering session duration, making the strategy a

time-efficient option. Whether findings may be related to fiber type-specific differences between muscles requires further investigation.

It is pertinent to note that participants expressed varying levels of discomfort while holding the stretch in both straight- and bent-leg positions. Although we did not employ a perceptual assessment, anecdotally the discomfort was greater overall in STRETCH than TRAD across the study period. Thus, long-term adherence to the inclusion of loaded inter-set stretch in RT programs may be dependent on an individual's ability to tolerate the heightened discomfort.

## Acknowledgments

We are grateful for the help of the following research assistants in conducting data collection: Roberto Arias, Ericka Johnson, Benjiman Mendelovits and Francesca Augustin.

## Author Contributions

**Conceptualization:** Andrew D. Vigotsky, Brad J. Schoenfeld.

**Data curation:** Derrick W. Van Every, Max Coleman, Andrew Alto.

**Formal analysis:** Andrew D. Vigotsky.

**Funding acquisition:** Brad J. Schoenfeld.

**Investigation:** Derrick W. Van Every, Max Coleman, Avery Rosa, Hugo Zambrano, Daniel Plotkin, Xavier Torres, Mariella Mercado, Douglas J. Oberlin.

**Methodology:** Eduardo O. De Souza, Andrew D. Vigotsky, Brad J. Schoenfeld.

**Project administration:** Derrick W. Van Every, Max Coleman, Douglas J. Oberlin, Brad J. Schoenfeld.

**Supervision:** Derrick W. Van Every, Max Coleman, Andrew Alto, Douglas J. Oberlin.

**Writing – original draft:** Brad J. Schoenfeld.

**Writing – review & editing:** Derrick W. Van Every, Max Coleman, Avery Rosa, Hugo Zambrano, Daniel Plotkin, Xavier Torres, Mariella Mercado, Eduardo O. De Souza, Andrew Alto, Douglas J. Oberlin, Andrew D. Vigotsky, Brad J. Schoenfeld.

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
