## [Decision Letter · Decision Letter 0]

7 Jun 2022

PONE-D-22-00944Loaded inter-set stretch may selectively enhance muscular adaptations of the plantar flexorsPLOS ONE

Dear Dr. Schoenfeld,

Thank you for submitting your manuscript to PLOS ONE. We regret that it has been so difficult obtaining a solid review for your paper, but feel confident that although it has taken far too long, we have some concrete reviews for your manuscript. However, after careful consideration, we feel while the paper certainly has merit and is likely to provide a good review for the literature, currently it does not yet meet PLOS ONE’s publication criteria as it currently stands. Therefore, we invite you to submit a revised version of the manuscript that addresses the points raised during the review process. Specifically, two expert Reviewers found your paper interesting but there were several areas that drew concern. In particular one of the Reviewers was concerned about your approach for statistical analysis. This was challenged as to some of the changes were very  modest regarding PF strength, and potential increases in muscle thickness due to the stretch intervention. Although this was identified as a trend, no trend statistics were presented to make the argument strongly convincing. The Reviewers have other specific comments that I hope that you find helpful in your revision.

We look forward to receiving your revised manuscript.

Kind regards,

Stephen E Alway, Ph.D.

Academic Editor

PLOS ONE

Journal Requirements:

2.Thank you for stating in your Funding Statement:

“This study was supported by a PSC CUNY grant from the State of New York received by BJS. Award # 63497-00 51. The funding agency played no role in study design, data collection, analysis, decision to publish, or preparation of the manuscript.”

3. Thank you for stating the following in the Competing Interests/Financial Disclosure* (delete as necessary) section:

“I have read the journal's policy and the authors of this manuscript have the following competing interests: BJS serves on the scientific advisory board of Tonal Corporation, a manufacturer of exercise equipment”

We note that one or more of the authors are employed by a commercial company: Tonal Corporation

5. Please include a caption for figure 3

Reviewers' comments:

Reviewer's Responses to Questions

**Comments to the Author**

1. Is the manuscript technically sound, and do the data support the conclusions?

Reviewer #1: Yes

Reviewer #2: Partly

2. Has the statistical analysis been performed appropriately and rigorously? 

Reviewer #1: N/A

Reviewer #2: No

3. Have the authors made all data underlying the findings in their manuscript fully available?

Reviewer #1: Yes

Reviewer #2: Yes

4. Is the manuscript presented in an intelligible fashion and written in standard English?

Reviewer #1: Yes

Reviewer #2: Yes

5. Review Comments to the Author

Reviewer #1: General comments: The paper is an intriguing one that is based on "older" work on the avian stretch model. I like the within-subject design; nice work.

Line 181, 199, and 213 (The figures on the PDF I have shows a bunch of question marks. I have no idea what it is supposed to show).

Otherwise, the design of the study is sound. The limitations mentioned in the paper are on point as well. If anything, this investigation shows how difficult it is to induce hypertrophy of the triceps surae. A study in trained bodybuilders would be interesting.

Reviewer #2: Thank you for the opportunity to review the manuscript titled “Loaded inter-set stretch may selectively enhance muscular adaptations of the plantar flexors.” (PONE-D-22-00944). The authors examined muscle thickness of the plantar flexor muscle group, and isometric plantar flexor strength after an 8-week training study which included resistance training, and resistance training with stretch between sets. The main finding was the soleus demonstrated greater response to the stretch protocol than the LG or MG.

General Comments:

The authors have presented an interesting study, however additional information around methods and interpretation of the results would enhance this paper. This reviewer finds some of the language used mis-conveys the findings and leads to ambiguity in the interpretation of the results, however I feel confident that with some added information and consideration of the below, the story will be strengthened.

The authors choice of the statistical evaluation of the data makes concrete interpretation of the results challenging, particularly in context of other stretch intervention literature. It is difficult to accept the changes noted as important when the authors are able to only suggest modest advantages to the stretch condition regarding PF strength, and potential increases in muscle thickness due to the stretch condition. While the authors state that the results are indicative of an important trend only, I’m not certain this is a strong enough argument to draw some of the conclusions made in this paper. For more specific comments please see below.

Specific Comments:

Abstract:

Line 2 – The word ‘longitudinal’ seems misplaced here (and in other places in the manuscript – Line 55 in the intro for one). Consider deleting as it refers more to the application of the intervention than the results.

Line 3/11/16 (and others) – The authors should consider their use of the word ‘hypertrophy’ in the context of this study. Although a change in muscle size (or thickness) was observed, I do not believe the measurements of true muscle hypertrophy exist in this methodology. Considering the literature, it appears this term is used often in context of stretch interventions, however I believe it is not accurate without a sample of muscle fibre size. At line 9 the authors refer to muscle thickness and this seems a more appropriate description of the outcome.

Introduction:

Line 29 – Simpson et al., (reference number 5) did not use the term ‘hypertrophy’ in their work, again as noted above, the authors should use caution about this in context of this paper. In order to justify this reference I recommend the authors change ‘hypertrophy’ to ‘ increases in muscle thickness’ as this is what was used and discussed in the Simpson et al. paper.

Line 60 – “…that the lateral gastrocnemius would experience greater hypertrophy than the other plantar flexors”. I assume the reference to the Schoenfeld et al. paper, and the work by Silva et al. are meant to serve as justification for as to why the authors believed the LG would show greater adaptation than the rest of the plantar flexors, however the introduction would be enhanced by an argument toward this end made by the authors (particularly when one of the main results relates to a difference between the LG, MG and Soleus).

Methods:

Line 64 – “recreationally active” males vs. “untrained” in Line 17 of the abstract. It is an important distinction when muscle growth is a key outcome.

Line 81 – Why did the authors choose to exclude resistance trained participants (or those that had trained in the past 6 months) particularly when Wadhi et al., and Silva et al. used trained individuals ( Line 36 and 45 of the Introduction). I suggest addition of justification around this point in the methods.

Line 96 – Is ‘stimulation’ the right word here? This conveys neuromuscular connotation. Consider a different word. Maybe ‘involvement’?

Line 97 – Describe the exercises used for this study. Considering these were loaded I would assume standard weight training machines were involved, but it would help evaluate the intervention to know which ones were used in this methodology. Even adding pictures if possible?

Line 98 – Why did the authors choose to use 2 days per week only? With the literature ranging from 1 day to 5 days per week it is worthwhile justifying this choice, particularly in regard to muscle growth.

Line 105 – What exercise was used for the establishment of the 10RM? Was this load kept the same between the seated and standing calf raises? Please add information around this.

Line 113 – How did the authors determine the set, rest and rep ranges, as well as the cadence of repetitions?

Line 116 – Given the literature varies extensively in the recommendations around stretch length and intensity, why did the authors choose a 20 second stretch (particularly when Silva et al. used 30?), and provide such a long rest interval between sets? I suggest this information is added to the paper.

Line 118 – For TRAD, when the subject was resting passively, were they in a loaded position still (ie-if they were performing standing calf raises, assuming the standard machine where weight is applied over the shoulders, did they step out from under the machine and rest standing? Seated? Or did they stay on the machine? For STRETCH, once the stretch was completed, the same question as the above. The authors might consider adding this information in the methods.

Line 126 – “Attempts were made…” does this suggest that not all loads were increased or that ‘attempts’ refers to the rep range? Suggest clarifying or deleting this as it seems to suggest not all participants underwent the same intervention.

Line 128 - Were participants allowed to perform aerobic/anaerobic cardiovascular-based exercises? Such as hill sprints, hurdles, explosive jumping, etc? Feasibly this could impact muscle growth as dramatically as resistance training (recognizing the authors suggested the subject pool was made up of recreationally active participants, I can’t imagine someone suddenly taking up hill sprints, but why make the distinction for resistance training only?)

Line 147 – The US measurements at 30% of the lower leg length needs qualification. At this point, the images are likely not across the widest part of the gastrocs and could be closer to the muscle tendon junction. I suggest the authors add information about this form of data collection as it is primary to their research question. Was each muscle group captured in an individual image at 30% of the lower leg length? If so, approximately where along the muscles was the measurement of depth taken? How did the image account for the difference in depth between the gastrocs and soleus? Was a different field of view used? Was the 30% position marked at the skin for all three muscles? What was the metric for when the quality of the image was ‘satisfactory’? How did the tech ‘obtain dimensions’ for the muscles? Given that even a small change in US probe orientation can produce a very different image, how did the authors/tech ensure the same spot measured pre and post intervention?

Line 147 – What was the position of the foot when the ultrasound images were being collected? How was the subject positioned? Please add this information

Line 151 – ‘Images for the MG and LG were measured as the distance from the superficial to deep aponeuroses…’ This doesn’t quite make sense. I interpret that this was the measure of muscle depth or thickness? If so, I suggest the authors replace the word ‘Images’ with what was actually measured here (muscle depth?) and then add information on how the measurement was done. If there was a software involved it would be ideal to have that information as well. Of particular importance is allowing the reader to understand if this is a depth measure (dropping a line from one aponeurosis to another?), how the same spot was measured each time (ie-did the authors define a measurement point a set distance from an identifiable landmark?). A figure showing these measurement techniques would be useful if possible.

Line 158 – ‘3 images were obtained for each site…’ With clarification of the above points it will become evident where the ‘sites’ are but the information at line 147 seems to indicate there is only one site being measured (30% of the lower leg length).

Line 163 – I suggest the authors include a bit more information around measurement of the MVC. Considering the contraction occurred for 5 seconds, was the maximal torque recorded at a certain point or averaged across the contraction (Line 174 – ‘the highest peak net joint moment…was used for analysis’ was this the highest toque achieved regardless of where in the contraction it occurred?).

Line 163 – Understanding the authors did not report voluntary activation, it may be worthwhile to report the instructions given to the participants as some were naïve to resistance training the contraction pre-intervention might be lower than post simply because the participants were not used to maximal contractions or were unable to recruit efficiently.

Line 173 – consider adding a reference after ‘increased performance’ due to visual feedback of the contraction (this one is good - https://doi.org/10.1139/apnm-2015-0639)

Statistical Analyses

Line 179 – The authors should include justification as to why they chose a hierarchical linear model over a linear regression for these data. Some discussion of variance or a suggestion as to what necessitated using this would help the reader understand the data slightly better. My interpretation of the references given (13 and 20) is they detail to the reader how these tests were done, not why. I believe the authors need to include the ‘why’ considering most of the stretch intervention literature uses different methods to assess similar data.

Line 228 – “…to avoid dichotomous interpretation of the results…” This sentence seems to suggest the authors are looking at trends, more than significant findings (ie-the goal is not to reject the null hypothesis it is to examine it). Which is important in its own way however the language used and conclusions drawn based on some of these findings are at odds with this statement. I encourage the authors to ensure the conclusions drawn are supported by these data and that by choosing the statistical methods they have, the authors may need to reconsider suggesting ‘change’ or ‘growth’ between muscles and conditions and instead suggest only trends. I am not convinced the findings are actually indicative of change but may simply be a trend in that direction (as I said, still important but needs to be carefully discussed).

Results:

Line 262 – This appears to be the only place the Volume Load measure appears in the paper (aside from Figure 2). I understand it likely links to the loading protocol described in the methods, but as it is not used in the discussion or to support the results, the authors might consider deleting it or using it for additional support of strength gains? (Same suggestion for the Figure)

Discussion:

Line 276 – 289 – This information may be better suited to the introduction and could aid in setting up the purpose statement and hypothesis more so than a description of the results.

Line 325 – “…may not reflect a global hypertrophying of the muscle.” I’m not sure what the authors mean here. Are the authors suggesting the protocol in this study created regional muscle growth? Is there additional evidence from this study to support this? It is an interesting idea but difficult to interpret from this statement. Consider rephrasing this.

Line 384 – the authors may consider the difference in findings comparing their untrained participants to those that were not naïve to resistance training. This seems to be a notable difference in this study and those the authors are drawing comparisons to!

Note – did the authors employ a measurement for the mobility of the ankle joint pre/post STRETCH intervention relative to TRAD? My though is perhaps some additional mobility could allow the ankle to plantar flex further in the eccentric phase of the calf-raise movement and facilitate greater utility of the muscle over the course of the protocol, thus enhancing size/strength. Just a thought here!

6. PLOS authors have the option to publish the peer review history of their article (what does this mean?). If published, this will include your full peer review and any attached files.

Reviewer #1: No

Reviewer #2: No

---

## [Author Response · Author response to Decision Letter 0]

16 Jun 2022

Reviewers' comments:

Reviewer's Responses to Questions

Comments to the Author

5. Review Comments to the Author

Reviewer #1: General comments: The paper is an intriguing one that is based on "older" work on the avian stretch model. I like the within-subject design; nice work.

AUTHOR RESPONSE: We appreciate your positive feedback and thank you for taking the time to review our paper. 

Line 181, 199, and 213 (The figures on the PDF I have shows a bunch of question marks. I have no idea what it is supposed to show).

Otherwise, the design of the study is sound. The limitations mentioned in the paper are on point as well. If anything, this investigation shows how difficult it is to induce hypertrophy of the triceps surae. A study in trained bodybuilders would be interesting.

AUTHOR RESPONSE: I believe the issue with the figures was because the editorial manager did not recognize the special characters in Word. We have revised and it appears this is now correct on the resubmission. 

Reviewer #2: Thank you for the opportunity to review the manuscript titled “Loaded inter-set stretch may selectively enhance muscular adaptations of the plantar flexors.” (PONE-D-22-00944). The authors examined muscle thickness of the plantar flexor muscle group, and isometric plantar flexor strength after an 8-week training study which included resistance training, and resistance training with stretch between sets. The main finding was the soleus demonstrated greater response to the stretch protocol than the LG or MG.

General Comments:

The authors have presented an interesting study, however additional information around methods and interpretation of the results would enhance this paper. This reviewer finds some of the language used mis-conveys the findings and leads to ambiguity in the interpretation of the results, however I feel confident that with some added information and consideration of the below, the story will be strengthened.

The authors choice of the statistical evaluation of the data makes concrete interpretation of the results challenging, particularly in context of other stretch intervention literature. It is difficult to accept the changes noted as important when the authors are able to only suggest modest advantages to the stretch condition regarding PF strength, and potential increases in muscle thickness due to the stretch condition. While the authors state that the results are indicative of an important trend only, I’m not certain this is a strong enough argument to draw some of the conclusions made in this paper. For more specific comments please see below.

AUTHOR RESPONSE: Thank you for your detailed comments; they have helped to improve the quality of our manuscript. We have addressed your comments on a point-by-point basis below, and made corresponding revisions in the manuscript highlighted in red text.

Specific Comments:

Abstract:

Line 2 – The word ‘longitudinal’ seems misplaced here (and in other places in the manuscript – Line 55 in the intro for one). Consider deleting as it refers more to the application of the intervention than the results.

AUTHOR RESPONSE: Fair point. We have deleted as requested.

Line 3/11/16 (and others) – The authors should consider their use of the word ‘hypertrophy’ in the context of this study. Although a change in muscle size (or thickness) was observed, I do not believe the measurements of true muscle hypertrophy exist in this methodology. Considering the literature, it appears this term is used often in context of stretch interventions, however I believe it is not accurate without a sample of muscle fibre size. At line 9 the authors refer to muscle thickness and this seems a more appropriate description of the outcome.

AUTHOR RESPONSE: We would note that muscle thickness is considered a valid measure of muscle size in the literature (see: https://www.frontiersin.org/articles/10.3389/fphys.2019.00247/full). That said, we understand your distinction here and thus have revised the paper to reflect changes in MT as opposed to hypertrophy where applicable. 

Introduction:

Line 29 – Simpson et al., (reference number 5) did not use the term ‘hypertrophy’ in their work, again as noted above, the authors should use caution about this in context of this paper. In order to justify this reference I recommend the authors change ‘hypertrophy’ to ‘ increases in muscle thickness’ as this is what was used and discussed in the Simpson et al. paper.

AUTHOR RESPONSE: Change made as requested.

Line 60 – “…that the lateral gastrocnemius would experience greater hypertrophy than the other plantar flexors”. I assume the reference to the Schoenfeld et al. paper, and the work by Silva et al. are meant to serve as justification for as to why the authors believed the LG would show greater adaptation than the rest of the plantar flexors, however the introduction would be enhanced by an argument toward this end made by the authors (particularly when one of the main results relates to a difference between the LG, MG and Soleus).

AUTHOR RESPONSE: Correct, we referenced the Schoenfeld et al. study as reason for our speculation. This was a secondary outcome of interest and thus, in trying to keep the introduction concise, we prefer to focus on the primary outcomes in this session. 

Methods:

Line 64 – “recreationally active” males vs. “untrained” in Line 17 of the abstract. It is an important distinction when muscle growth is a key outcome.

AUTHOR RESPONSE: We have revised to state “untrained but recreationally active” as the subjects did not participate in regimented resistance training

Line 81 – Why did the authors choose to exclude resistance trained participants (or those that had trained in the past 6 months) particularly when Wadhi et al., and Silva et al. used trained individuals ( Line 36 and 45 of the Introduction). I suggest addition of justification around this point in the methods.

AUTHOR RESPONSE: The study involved training just the calf muscles (without training the other lower limb muscles to control for potential confounding), and it would have been virtually impossible to get resistance trained subjects to give up training their lower body muscles for several months. Thus, by default we chose to employ untrained individuals for the sample.

Line 96 – Is ‘stimulation’ the right word here? This conveys neuromuscular connotation. Consider a different word. Maybe ‘involvement’?

AUTHOR RESPONSE: Changed as requested

Line 97 – Describe the exercises used for this study. Considering these were loaded I would assume standard weight training machines were involved, but it would help evaluate the intervention to know which ones were used in this methodology. Even adding pictures if possible?

AUTHOR RESPONSE: Good point. We have provided a description of the equipment. 

Line 98 – Why did the authors choose to use 2 days per week only? With the literature ranging from 1 day to 5 days per week it is worthwhile justifying this choice, particularly in regard to muscle growth.

AUTHOR RESPONSE: Two days per week appears to be sufficient for optimizing results in hypertrophy-oriented programs (https://pubmed.ncbi.nlm.nih.gov/30558493/); there is not good evidence that additional frequencies promote better results.

Line 105 – What exercise was used for the establishment of the 10RM? Was this load kept the same between the seated and standing calf raises? Please add information around this.

AUTHOR RESPONSE: Good catch. We have added the info as requested.

Line 113 – How did the authors determine the set, rest and rep ranges, as well as the cadence of repetitions?

AUTHOR RESPONSE: The program aligns with typical hypertrophy-oriented RT programs and is consistent with general guidelines in the literature (see: https://pubmed.ncbi.nlm.nih.gov/27433992/;
https://pubmed.ncbi.nlm.nih.gov/30558493/;
https://pubmed.ncbi.nlm.nih.gov/33671664/) 

Line 116 – Given the literature varies extensively in the recommendations around stretch length and intensity, why did the authors choose a 20 second stretch (particularly when Silva et al. used 30?), and provide such a long rest interval between sets? I suggest this information is added to the paper.

AUTHOR RESPONSE: There are no guidelines as yet as to optimal inter-set stretch duration. We carried out pilot testing and determined that subjects generally started to experience high levels of discomfort after 20 seconds and thus decided to employ this duration. We do in fact note in our limitations section that other durations may have provided an alternative outcome and thus this area requires future research. 

Line 118 – For TRAD, when the subject was resting passively, were they in a loaded position still (ie-if they were performing standing calf raises, assuming the standard machine where weight is applied over the shoulders, did they step out from under the machine and rest standing? Seated? Or did they stay on the machine? For STRETCH, once the stretch was completed, the same question as the above. The authors might consider adding this information in the methods.

AUTHOR RESPONSE: The straight leg calf raise was performed in a leg press. All subjects rested passively in the respective units during the rest periods, but were in an unloaded state so this had no bearing on results. 

Line 126 – “Attempts were made…” does this suggest that not all loads were increased or that ‘attempts’ refers to the rep range? Suggest clarifying or deleting this as it seems to suggest not all participants underwent the same intervention.

AUTHOR RESPONSE: Change made as requested.

Line 128 - Were participants allowed to perform aerobic/anaerobic cardiovascular-based exercises? Such as hill sprints, hurdles, explosive jumping, etc? Feasibly this could impact muscle growth as dramatically as resistance training (recognizing the authors suggested the subject pool was made up of recreationally active participants, I can’t imagine someone suddenly taking up hill sprints, but why make the distinction for resistance training only?)

AUTHOR RESPONSE: We did not make the distinction for aerobic activities, but we asked about their general activity levels and none of the subjects reported carrying out intense anaerobic exercise

Line 147 – The US measurements at 30% of the lower leg length needs qualification. At this point, the images are likely not across the widest part of the gastrocs and could be closer to the muscle tendon junction. I suggest the authors add information about this form of data collection as it is primary to their research question. Was each muscle group captured in an individual image at 30% of the lower leg length? If so, approximately where along the muscles was the measurement of depth taken? How did the image account for the difference in depth between the gastrocs and soleus? Was a different field of view used? Was the 30% position marked at the skin for all three muscles? What was the metric for when the quality of the image was ‘satisfactory’? How did the tech ‘obtain dimensions’ for the muscles? Given that even a small change in US probe orientation can produce a very different image, how did the authors/tech ensure the same spot measured pre and post intervention?

AUTHOR RESPONSE: The 30% measure was within the level of the muscle belly; we have done extensive experimentation in this regard and within 25% to 30% seems to provide the most consistent evaluation of the belly of the muscle. As noted in the methods, we employed a range of frequencies to evaluate the muscle based on interindividual differences. We have added information to indicate that measurements were taken using a tape measure and that sites were marked with a felt-tip pen. Our ICC’s are ~0.99 for measurement of the respective muscles, indicating a very high inter-session reliability. 

Line 147 – What was the position of the foot when the ultrasound images were being collected? How was the subject positioned? Please add this information

AUTHOR RESPONSE: The ankle was held in neutral position; this info has been added to the manuscript. 

Line 151 – ‘Images for the MG and LG were measured as the distance from the superficial to deep aponeuroses…’ This doesn’t quite make sense. I interpret that this was the measure of muscle depth or thickness? If so, I suggest the authors replace the word ‘Images’ with what was actually measured here (muscle depth?) and then add information on how the measurement was done. If there was a software involved it would be ideal to have that information as well. Of particular importance is allowing the reader to understand if this is a depth measure (dropping a line from one aponeurosis to another?), how the same spot was measured each time (ie-did the authors define a measurement point a set distance from an identifiable landmark?). A figure showing these measurement techniques would be useful if possible.

AUTHOR RESPONSE: We have revised for clarity. We also have added a representative image to illustrate the measurements.

Line 158 – ‘3 images were obtained for each site…’ With clarification of the above points it will become evident where the ‘sites’ are but the information at line 147 seems to indicate there is only one site being measured (30% of the lower leg length).

AUTHOR RESPONSE: We evaluated both the lateral and medial gastrocnemius so there were 2 sites along the width of the muscle; we have clarified this in the text.

Line 163 – I suggest the authors include a bit more information around measurement of the MVC. Considering the contraction occurred for 5 seconds, was the maximal torque recorded at a certain point or averaged across the contraction (Line 174 – ‘the highest peak net joint moment…was used for analysis’ was this the highest toque achieved regardless of where in the contraction it occurred?).

AUTHOR RESPONSE: Maximal torque was determined by the unit as the highest torque at any point during the MVC. This is reflected in the sentence: “The highest peak net joint moment from each of the 4 trials for each position was used for analysis”

Line 163 – Understanding the authors did not report voluntary activation, it may be worthwhile to report the instructions given to the participants as some were naïve to resistance training the contraction pre-intervention might be lower than post simply because the participants were not used to maximal contractions or were unable to recruit efficiently.

AUTHOR RESPONSE: Good suggestion. We have included a sentence to describe the instructions as well as the fact that participants were provided with a practice trial prior to testing.

Line 173 – consider adding a reference after ‘increased performance’ due to visual feedback of the contraction (this one is good - https://doi.org/10.1139/apnm-2015-0639)

AUTHOR RESPONSE: Thank you for the suggestion. We have included the reference as requested.

Statistical Analyses

Line 179 – The authors should include justification as to why they chose a hierarchical linear model over a linear regression for these data. Some discussion of variance or a suggestion as to what necessitated using this would help the reader understand the data slightly better. My interpretation of the references given (13 and 20) is they detail to the reader how these tests were done, not why. I believe the authors need to include the ‘why’ considering most of the stretch intervention literature uses different methods to assess similar data.

AUTHOR RESPONSE: We’ve added text to elaborate that these models are indicated due to the within-subject nature of our study. In principle, one could use OLS regression like they do in econometrics (treating subjects as “fixed-effects”), but this has some interpretation issues. We’ve also provided some explanation linking our models to linear mixed-effects models, with which some readers may be more familiar. 

Line 228 – “…to avoid dichotomous interpretation of the results…” This sentence seems to suggest the authors are looking at trends, more than significant findings (ie-the goal is not to reject the null hypothesis it is to examine it). Which is important in its own way however the language used and conclusions drawn based on some of these findings are at odds with this statement. I encourage the authors to ensure the conclusions drawn are supported by these data and that by choosing the statistical methods they have, the authors may need to reconsider suggesting ‘change’ or ‘growth’ between muscles and conditions and instead suggest only trends. I am not convinced the findings are actually indicative of change but may simply be a trend in that direction (as I said, still important but needs to be carefully discussed).

AUTHOR RESPONSE: To hopefully address your concerns more formally, we are unsure what you mean by “trend,” but our rationale is grounded in statistics and decision theory. Significance testing is designed to be a decision process, and since drawing knowledge or inferences from studies is not a decision process (see Section 3, especially 3.2, in Greenland 2017), we are not interested in testing the (or a specific) null hypothesis. Rather, our primary interest is to obtain estimates of the effects along with the uncertainty of those estimates. The CI provides us with a range of values that are compatible with our data—to this end, we are quite careful to discuss the implications of the full range of values contained by the CI. After all, the lower bound of the CI is just as compatible with the data as the upper bound of the CI. Thus, we think it’s important for readers to appreciate all potential implications of our data rather than making dichotomous decisions based on whether the CI crosses zero.

Some references that expand on this viewpoint:

• Greenland, S. (2017). Invited Commentary: The Need for Cognitive Science in Methodology. American Journal of Epidemiology, 186(6), 639-645.

• Amrhein, V., Greenland, S., & McShane, B. (2019). Scientists rise up against statistical significance. Nature, 567(7748), 305-307.

• Greenland, S. (2021). Analysis goals, error‐cost sensitivity, and analysis hacking: Essential considerations in hypothesis testing and multiple comparisons. Paediatric and Perinatal Epidemiology, 35(1), 8-23.

We have tried to be cautious in our interpretation of findings throughout the manuscript when attempting to draw evidence-based conclusions. If there are specific instances where you believe we’ve overinterpreted the data, please let us know and we’d be happy to adjust our language or discuss as needed.

Results:

Line 262 – This appears to be the only place the Volume Load measure appears in the paper (aside from Figure 2). I understand it likely links to the loading protocol described in the methods, but as it is not used in the discussion or to support the results, the authors might consider deleting it or using it for additional support of strength gains? (Same suggestion for the Figure)

AUTHOR RESPONSE: Fair point. We have deleted the volume load data and removed its inclusion in the figure. 

Discussion:

Line 276 – 289 – This information may be better suited to the introduction and could aid in setting up the purpose statement and hypothesis more so than a description of the results.

AUTHOR RESPONSE: We did cover these topics in the intro. We are simply reiterating the info to set up the discussion of results, so we feel it is better suited here as it would be rather redundant in the intro.

Line 325 – “…may not reflect a global hypertrophying of the muscle.” I’m not sure what the authors mean here. Are the authors suggesting the protocol in this study created regional muscle growth? Is there additional evidence from this study to support this? It is an interesting idea but difficult to interpret from this statement. Consider rephrasing this.

AUTHOR RESPONSE: We agree this was a bit ambiguous. We have revised for clarity. Thanks.

Line 384 – the authors may consider the difference in findings comparing their untrained participants to those that were not naïve to resistance training. This seems to be a notable difference in this study and those the authors are drawing comparisons to!

AUTHOR RESPONSE: Fair point. We have added a sentence to reflect potential discrepancies between trained and untrained participants.

Note – did the authors employ a measurement for the mobility of the ankle joint pre/post STRETCH intervention relative to TRAD? My though is perhaps some additional mobility could allow the ankle to plantar flex further in the eccentric phase of the calf-raise movement and facilitate greater utility of the muscle over the course of the protocol, thus enhancing size/strength. Just a thought here!

AUTHOR RESPONSE: This is an interesting hypothesis, as it is possible the intervention may have had additional flexibility benefits. However, this was not a predetermined outcome and thus unfortunately we did not measure ROM changes.

---

## [Decision Letter · Decision Letter 1]

9 Aug 2022

Loaded inter-set stretch may selectively enhance muscular adaptations of the plantar flexors

PONE-D-22-00944R1

Dear Dr. Schoenfeld,

After careful review of your revision, we are pleased to inform you that your manuscript has been judged scientifically suitable for publication and will be formally accepted for publication once it meets all outstanding technical requirements.

Kind regards,

Stephen E Alway, Ph.D.

Academic Editor

PLOS ONE

Additional Editor Comments (optional):

Reviewers' comments:

Reviewer's Responses to Questions

**Comments to the Author**

1. If the authors have adequately addressed your comments raised in a previous round of review and you feel that this manuscript is now acceptable for publication, you may indicate that here to bypass the “Comments to the Author” section, enter your conflict of interest statement in the “Confidential to Editor” section, and submit your "Accept" recommendation.

Reviewer #2: All comments have been addressed

2. Is the manuscript technically sound, and do the data support the conclusions?

Reviewer #2: Yes

3. Has the statistical analysis been performed appropriately and rigorously? 

Reviewer #2: Yes

4. Have the authors made all data underlying the findings in their manuscript fully available?

Reviewer #2: Yes

5. Is the manuscript presented in an intelligible fashion and written in standard English?

Reviewer #2: Yes

6. Review Comments to the Author

Reviewer #2: Thank you for your thorough explanations and thoughtful revisions to the paper. I am interested to see where this work will go from here, and additional studies to come from it.

7. PLOS authors have the option to publish the peer review history of their article (what does this mean?). If published, this will include your full peer review and any attached files.

Reviewer #2: No

---

## [Editor Report · Acceptance letter]

16 Aug 2022

PONE-D-22-00944R1 

Loaded inter-set stretch may selectively enhance muscular adaptations of the plantar flexors 

Dear Dr. Schoenfeld:

I'm pleased to inform you that your manuscript has been deemed suitable for publication in PLOS ONE. Congratulations! Your manuscript is now with our production department. 

Kind regards, 

on behalf of

Dr. Stephen E Alway 

Academic Editor

PLOS ONE